# Somatic Mutation Profiling in the Liquid Biopsy and Clinical Analysis of Hereditary and Familial Pancreatic Cancer Cases Reveals *KRAS* Negativity and a Longer Overall Survival

**DOI:** 10.3390/cancers13071612

**Published:** 2021-03-31

**Authors:** Julie Earl, Emma Barreto, María E. Castillo, Raquel Fuentes, Mercedes Rodríguez-Garrote, Reyes Ferreiro, Pablo Reguera, Gloria Muñoz, David Garcia-Seisdedos, Jorge Villalón López, Bruno Sainz, Nuria Malats, Alfredo Carrato

**Affiliations:** 1Molecular Epidemiology and Predictive Tumor Markers Group, Ramón y Cajal Health Research Institute (IRYCIS), Carretera Colmenar Km 9100, 28034 Madrid, Spain; emma.barreto@salud.madrid.org (E.B.); marien.castillo@salud.madrid.org (M.E.C.); mercedes.rodrí guez@salud.madrid.org (M.R.-G.); reyes-ferreiro@hotmail.com (R.F.); pablo.reguera@salud.madrid.org (P.R.); jorge.villalon@salud.madrid.org (J.V.L.); alfredo.carrato@salud.madrid.org (A.C.); 2Biomedical Research Network in Cancer (CIBERONC), C/Monforte de Lemos 3-5. Pabellón 11, 28029 Madrid, Spain; nmalats@cnio.es; 3Translational Genomics Core Facility, Ramón y Cajal Health Research Institute (IRYCIS), 28034 Madrid, Spain; mariagloria.munoz@salud.madrid.org (G.M.); dgarcia@ebi.ac.uk (D.G.-S.); 4Department of Biochemistry, Universidad Autónoma de Madrid (UAM), Ramón y Cajal Health Research Institute (IRYCIS), 28034 Madrid, Spain; bsainz@iib.uam.es; 5Instituto de Investigaciones Biomédicas “Alberto Sols” (IIBM), CSIC-UAM, C/Arzobispo Morcillo, 4, 28029 Madrid, Spain; 6Cancer Stem Cell and Fibroinflammatory Group, Chronic Diseases and Cancer, Area 3- IRYCIS, 28029 Madrid, Spain; 7Genetic and Molecular Epidemiology Group, Spanish National Cancer Research Centre (CNIO), 28029 Madrid, Spain; 8Department of Medicine and Medical Specialties, Medicine Faculty, Alcala University, Plaza de San Diego, s/n, 28801 Alcalá de Henares, Spain

**Keywords:** liquid biopsy, cfDNA, hereditary and familial pancreatic cancer, somatic mutation profiling, potentially druggable genes

## Abstract

**Simple Summary:**

Pancreatic ductal adenocarcinoma (PDAC) has a poor prognosis. *KRAS* mutations occur in up to 95% of cases and render the tumor resistant to many types of therapy. Therefore, these patients are treated with traditional cytotoxic agents, according to guidelines. The familial or hereditary form of the disease accounts for up to 10–15% of cases. We hypothesized that hereditary and Familial Pancreatic Cancer cases (H/FPC) have a distinct tumor specific mutation profile due to the presence of pathogenic germline mutations and we used circulating free DNA (cfDNA) in plasma to assess this hypothesis. H/FPC cases were mainly *KRAS* mutation negative and harbored tumor specific mutations that are potential treatment targets in the clinic. Thus, we conclude that cases with a hereditary or familial background can be treated with newer and more effective agents that may ultimately improve their overall survival.

**Abstract:**

Pancreatic ductal adenocarcinoma (PDAC) presents many challenges in the clinic and there are many areas for improvement in diagnostics and patient management. The five-year survival rate is around 7.2% as the majority of patients present with advanced disease at diagnosis that is treatment resistant. Approximately 10–15% of PDAC cases have a hereditary basis or Familial Pancreatic Cancer (FPC). Here we demonstrate the use of circulating free DNA (cfDNA) in plasma as a prognostic biomarker in PDAC. The levels of cfDNA correlated with disease status, disease stage, and overall survival. Furthermore, we show for the first time via BEAMing that the majority of hereditary or familial PDAC cases (around 84%) are negative for a *KRAS* somatic mutation. In addition, *KRAS* mutation negative cases harbor somatic mutations in potentially druggable genes such as *KIT, PDGFR, MET, BRAF,* and *PIK3CA* that could be exploited in the clinic. Finally, familial or hereditary cases have a longer overall survival compared to sporadic cases (10.2 vs. 21.7 months, respectively). Currently, all patients are treated the same in the clinic with cytotoxic agents, although here we demonstrate that there are different subtypes of tumors at the genetic level that could pave the way to personalized treatment.

## 1. Introduction

The incidence and mortality rates of adenocarcinoma of the pancreas (PDAC) are almost equal [1]. Today, PDAC is the third leading cause of cancer death in the EU and by 2030 it is projected to be the second leading cause of cancer-related death [2], surpassing breast, prostate, and colorectal cancer. The overall survival at 5 years is around 7.2% due to the fact that the majority of patients present with advanced and treatment resistant disease at diagnosis. *KRAS* somatic mutations are present in around 90% of sporadic tumors and *TP53, CDKN2A,* and *SMAD4* mutations are also commonly found. However, none of these somatic changes are druggable at present time. In fact, PDAC tumors have an average of 63 somatic alterations that affect different signaling pathways [3]. The best described common risk factors for sporadic PDAC include tobacco, alcohol, diabetes, chronic pancreatitis, and obesity [4]. Family history is also an important risk factor and approximately 10% of PDAC cases have a hereditary or familial basis [5]. Hereditary pancreatic cancer are associated with a known cancer syndrome such as hereditary breast–ovarian cancer (HBOC), Peutz–Jeghers (PJ), Hereditary Pancreatitis (HP), Familial Atypical Multiple Mole Melanoma (FAMMM), and Lynch syndrome and harbor germline pathogenic mutations in genes such as *BRCA2, MLH1,* and *CDKN2A* [6]. Whereas, Familial Pancreatic Cancer (FPC) is defined as a family with at least one pair of affected first degree relatives with no identified genetic basis and account for 4–10% of PDAC patients [7,8]. The Spanish familial pancreatic cancer registry (PANGENFAM) was established in 2009 with the principal objective to characterize the phenotypic and genetic background of FPC [9].

Specific, sensitive and minimally invasive biomarkers are needed in order to accurately diagnose PDAC at a potentially curable stage and aid in patient management during treatment. CA19-9 is currently used in the clinic, although the sensitivity and specificity for the diagnosis of symptomatic PDAC is 79–81% and 82–90%, respectively [10]. Several potential biomarkers have been recently described such as a three-protein panel in urine [11], Galectin-1 (Gal-1) in serum [12], thrombospondin-2 (THBS2) in plasma [13], circulating tumor DNA (ctDNA) [14], and the IMMray™ PanCan-d 29 biomarker signature in serum [15]. The term the “liquid biopsy” was coined in 2010 [16], and is defined as the detection and analysis of molecules (e.g., protein, DNA, RNA), cells or extracellular vesicles (e.g., exosomes) that originate from the primary tumor in blood and other body fluids, such as saliva, cerebrospinal fluid, and feces. Since fresh tissue, in the form of biopsies, is scarce and prohibitive for many PDAC patients, liquid biopsies represent an attractive surrogate system to provide essential information regarding diagnosis, stage, etc. Likewise, there is an important and inherent degree of heterogeneity in primary PDAC tumors and associated metastatic lesions [17], which can only be studied in depth via liquid biopsies due to the shortage of primary and metastatic tumor tissue. Thus, research to maximize the potential information that a liquid biopsy can offer has exploded in the past decade. cfDNA, which consists of double-stranded DNA molecules of 70 to 200 bp [18,19], is released into the blood stream by apoptotic or necrotic cells or in extracellular vesicles such as exosomes. cfDNA is present in all individuals, although levels up 40 times higher are detected in patients with tumors or inflammatory disease [20]. In fact, cfDNA detection has been used previously as a prognostic and predictive marker in pancreatic cancer [21,22].

This study aimed to analyze the use of cfDNA as a prognostic and predictive marker in PDAC. Furthermore, the somatic mutation profile of Familial or Hereditary PDAC (H/FPC) and sporadic PDAC cases was also analyzed.

## 2. Materials and Methods

### 2.1. Identification and Classification of Patients

The study was approved by the local ethics committee and all patients signed the associated informed consent. PDAC patients were identified in the medical oncology and general surgery departments of the Ramon y Cajal University Hospital in Madrid, Spain. Cases that complied with the inclusion criteria of the Spanish Familial Pancreatic cancer registry (PANGENFAM) were classified as Hereditary or FPC (H/FPC cohort) [9] and included cases with and without a known genetic cause. Cases with no reported hereditary or familial pancreatic cancer syndrome were classified as sporadic cases (SP cohort). High-risk family members of H/FPC cases in the secondary screening program that were diagnosed with pancreatic cysts or intraductal papillary mucinous neoplasm (IPMN) were also included and one case that was initially included in the study as a possible PDAC, but was later confirmed as an IPMN (pancreatic cysts cohort). Blood samples in EDTA tubes were taken on entry into the study and plasma was extracted and stored until cfDNA extraction and somatic mutation analysis. All clinical and personal data was stored in a secure database Research Electronic Data Capture (REDCap: https://www.project-redcap.org/).

### 2.2. Isolation of cfDNA from Plasma

A total of two different methods of cfDNA isolation were compared to determine the most appropriate method for PCR and sequencing based analysis of cfDNA, the Maxwell^®^ RSC Instrument and the QIAamp circulating nucleic acid kit (Qiagen, Venlo, Netherlands). DNA was extracted from 1 mL of plasma using the Maxwell^®^ RSC Instrument (Promega, Madison, WI, USA) and eluted in a final volume of 50 μL from 136 patients diagnosed with PDAC and 29 individuals diagnosed with pancreatic cysts and IPMN. Plasma samples were also obtained from 40 healthy individuals with no known history of digestive disease or cancer, provided by the BioBank Hospital Ramón y Cajal-IRYCIS (PT13/0010/0002), integrated in the Spanish National Biobanks Network (ISCIII Biobank Register No. B.0000678, Spain). cfDNA was isolated as 10 plasma pools that consisted of 4 individuals in each. The concentration of cfDNA was determined in all samples using the QuantiFluor^®^ dsDNA System (Promega, Madison, WI, USA) kit and analyzed using the Quantus Fluorometer (Promega, Madison, WI, USA).

### 2.3. Detection of Mutant KRAS in Plasma by BEAMing

The presence of a mutation in *KRAS* codons 12, 13, 59, 61, 117, and 146 was determined in cfDNA isolated using the Maxwell^®^ RSC system and the OncoBEAM KRAS CRC kit (Sysmex Inostics, Hamburg, Germany) using the BEAMing technology (Sysmex Inostics, Hamburg, Germany) according to the manufacturer’s instructions, which also includes positive and negative assay controls.

### 2.4. Sequencing of cfDNA Using the TruSight15 System (Illumina)

Cell free DNA was extracted from 1–3 mL of plasma using the QIAamp circulating nucleic acid kit (Qiagen, Venlo, Netherlands) according to the manufacturer’s instructions. The DNA preparation obtained was purified using the Agentcourt AMPure XP Reagent (Beckman Coulter, Brea, CA, USA) in two successive steps in order to isolate cfDNA of approximately 160–170 bp. DNA was added to the Agentcourt AMPure XP Reagent at a ratio of 0.7× and then at a ratio of 2× the initial volume of DNA. The sample was then vortexed, centrifuged and incubated for 5 min at room temperature and then placed in a magnetic rack and incubated for 5 min. The supernatant containing the cfDNA was placed in a new Eppendorf and more Agentcourt AMPure XP Reagent was added for the second round of purification. The same steps were repeated and the supernatant was finally discarded. Then, 70% ethanol was added to the sample and then vortexed, centrifuged, and placed in the magnetic rack for 30 s. The supernatant was removed and the sample was washed with 70% ethanol in order to obtain a dry pellet. Finally, nuclease free water was added and the sample was vortexed, centrifuged, incubated at room temperature for 2 min and then incubated in the magnetic rack for 2 min. The purified cfDNA was analyzed with the Tape Station 2200 using the HS D1000 kit (Agilent Technologies, Santa Clara, CA, USA) to confirm the presence of a single band of approximately 160 bp. After quantification using the Qubit fluorimeter 2.0 (ThermoFisher Scientific, Waltham, MA, USA) with the HS DNA, the cfDNA was diluted to 2 ng/μL to prepare genome libraries using the TruSight Tumor 15 kit (Illumina, San Diego, CA, USA), which is optimized for low DNA input.

The sequencing panel included the genes *AKT1, BRAF, EGFR, ERBB2, FOXL2, GNA11, GNAQ, KIT, KRAS, MET, NRAS, PDGFRA, PIK3CA, RET,* and *TP53.* In total, 27 PDAC cases (20 H/FPC cases and 7 sporadic cases) were included in the panel sequencing analysis of cfDNA, and five of these samples were performed in duplicate. Then, two healthy individuals, one case of previous breast cancer and one previous pancreatic neuroendocrine tumor case, were also included in the study as control samples.

### 2.5. Identification of Pathogenic Somatic Variants

The BaseSpace Variant Interpreter (https://variantinterpreter.informatics.illumina.com/home, last accessed date: 1 July 2020) from Illumina was used for the identification of pathogenic somatic variants, which was specifically designed to analyze the sequencing data generated using the TruSight 15 kit. A detailed description of the analysis pipeline is shown in Appendix A. Briefly, the variant call files (vcf) were uploaded and the “Small Variant Consequences” filter was applied that included stop gain, stop loss, splice site, missense, frameshift, deletions, insertions, and initiator codon (ATG) loss. Those variants with a deleterious, probably or possibly damaging consequence according to the SIFT and PolyPhen parameters were retained. The frequency of somatic variants was analyzed using the Catalogue of Somatic Mutations in Cancer public database (COSMIC: (https://cancer.sanger.ac.uk/cosmic, last accessed date: 1 July 2020). Samples that did not reach an average minimum amplicon coverage of 500 were excluded, and variants that passed the quality filters of genotype quality, variant frequency, and strand bias were retained.

### 2.6. Statistical Analysis

The R program 3.4.3 was used for statistical analysis. The Mann–Whitney test was used to analyze the differences in concentration of cfDNA between the different groups, according to disease status and stage. The Chi square test was used to study the differences in the frequency of somatic mutations in the different groups. A One-Way ANOVA test was used to determine the differences in age at diagnosis and the Fishers exact test was used to analyze the differences between sex and disease stage distributions in each group.

The “survival” package of R was used to perform survival analysis. Overall survival (OS) was determined from the date of diagnosis until the date of death (event) or the date of the last follow-up (censored). Patients were divided into 3 groups according to total cfDNA concentration for survival analysis: high (≥3rd quartile), medium (≤3rd quartile and ≥1st quartile), and low (≤1st quartile). The Kaplan Meyer and Log Rank test were used to analyze overall survival among different subgroups of patient according to cfDNA levels and classification as sporadic or H/FPC cases. The analysis was adjusted for age and disease stage at diagnosis, sex and 1st line treatment and the corresponding hazard ratios were calculated.

## 3. Results

### 3.1. Patient Characteristics

In total, 184 individuals were recruited in the entire study, including 145 cases (102 sporadic and 43 familial or hereditary PDAC cases), 29 patients with pancreatic cysts, and 40 healthy controls. The demographic characteristics of the individuals included in the study are summarized Table 1 and more detailed information of the clinical characteristics of the different cohorts is provided in Appendix A. The PDAC cohort included 71 males and 74 females with a median age of 69 years (29–90 years). The median age at diagnosis of sporadic cases (SP) was 70 years and 65.5 years for Hereditary or Familial PDAC cases (H/FPC), the difference was statistically significant according to the one-way ANOVA test (*p* = 0.000113). The Male:Female ratio for sporadic cases was 0.89 and 0.65 and this difference was not statistically significant according to the Fishers exact test. The pancreatic cyst cohort consisted of 10 males and 19 females with a median age of 53 (37–81 years) and four patients in this cohort finally underwent a surgical resection of the pancreatic lesion due to a suspicion of malignancy (Appendix A). The 40 healthy individual cohort consisted of 17 females and 23 males with a median age of 39 years (18–63 years) and no known digestive diseases.

A total of 136 PDAC patients (94 sporadic and 42 H/FPC) were included in the cfDNA quantification analysis, 54 in the *KRAS* detection analysis in cfDNA via BEAMing (23 sporadic and 31 H/FPC) and 20 H/FPC cases and 7 sporadic cases were included in the panel sequencing analysis of cfDNA. There were no significant differences between the age, sex, and disease stage at diagnosis distribution of the cohorts used for BEAMing and sequencing analysis compared with the entire cohort.

### 3.2. Correlation of cfDNA Levels with Clinical Parameters

The Maxwell^®^ RSC kit favors the isolation of small DNA fragments within the expected cfDNA size range of 150 to 200 bp with a low contamination of high-molecular weight genomic DNA (Appendix A). Thus, this method was used for the study of total cfDNA analysis in 1 mL of plasma and also for the BEAMing-based studies to avoid the amplification of non-tumor genomic DNA. The total cfDNA concentration in plasma (ng/μL) was determined for 136 patients with PDAC, 29 with pancreatic cysts and 40 healthy individuals (Figure 1a). The median cfDNA level in healthy individuals was 0.01 ng (0.005–0.09), 0.03 ng (0.005–0.7) in patients with pancreatic cysts, 0.0575 ng (0.01–4.17) in H/FPC PDAC cases, and 0.07 ng (0.005–2.055) in sporadic PDAC cases. There were significantly higher levels of cfDNA in patients with pancreatic cysts (*p* = 0.021), sporadic PDAC (*p* ≤ 0.001) and H/FPC PDAC (*p* ≤ 0.001) compared to healthy individuals. Furthermore, there was significantly higher levels of cfDNA levels in both H/FPC PDAC (*p* = 0.02314) and sporadic PDAC cases (*p* = 0.01374) compared to patients with pancreatic cysts. The median cfDNA concentration in all PDAC cases was 0.0675 ng (0.005–4.17) and there was no significant difference in cfDNA levels between H/FPC PDAC and sporadic PDAC; 0.0575 vs. 0.07 ng, respectively. The median cfDNA level in resectable cases was 0.0575 ng (0.0050–2.0000), 0.0675 ng (0.0350–2.0000) in locally advanced, and 0.07 ng (0.005–4.000) in metastatic cases (Appendix A). Even though the median cfDNA level increased according to disease stage, the difference did not reach statistical significance. However, the levels of cfDNA were significantly higher in resectable cases (*p* ≤ 0.001), locally advanced (*p* ≤ 0.001), and metastatic cases (*p* ≤ 0.001) compared to healthy controls. Furthermore, the cfDNA levels were also significantly higher in locally advanced (*p* = 0.02) and metastatic cases (*p* = 0.02) compared to patients with pancreatic cysts. There was no significant association between cfDNA levels and tumor size according to the Pearson Correlation (−0.1).

Survival and cfDNA total concentration data were available for 134 PDAC cases. Cases were followed up for a median of 12 months (0.4–60 months) and survival analysis was performed based on total cfDNA levels and censored at 5 years. Patients were classified into 3 groups: high (>0.2037 ng), medium (>0.035 ng and <0.2037 ng), and low (<0.035 ng) cfDNA levels at diagnosis. The concentration of cfDNA in plasma significantly correlated with overall survival (OS) and patients with a high cfDNA concentration at diagnosis (>0.2037 ng) had a significantly shorter OS, with a median overall survival of 8.2 vs. 11.4 and 15.8 months for medium and low levels, respectively (Figure 1b). The Hazard Ratios for a low cfDNA level at diagnosis was 0.6 (*p* = 0.04), 0.5 for a medium level (*p* = 0.01) and 1.8 (*p* = 0.04) for a high level (Appendix A). Furthermore, cfDNA levels were determined in plasma before and 1 month after a surgical resection of the primary tumor. There was a significant reduction in the cfDNA concentration from 0.11 ng (0.025–5.5 ng) before surgery to 0.025 ng (0.01–1.25 ng) after surgery (*p* = 0.0024) (Appendix A), which supports the idea that cfDNA levels are indicative of tumor burden.

### 3.3. Analysis of Somatic Mutations in Plasma

BEAMing was performed with cfDNA isolated from 1 mL of plasma using the Maxwell^®^ RSC kit from 54 PDAC cases, which included 31 H/FPC cases and 23 sporadic cases. The frequency of *KRAS* mutations in codon 12 and 13 was 70% in sporadic cases and 16% in familial cases (Figure 2a,b), which was statistically significant (*p* ≤ 0.001), and indicated that *KRAS* somatic mutations are less frequent in H/FPC cases compared to sporadic cases. According to disease stage, KRAS positivity in sporadic and H/FPC cases was 67 vs. 17% for locally advanced cases and 75 vs. 17% in metastatic cases, respectively, which was statistically significant (*p* ≤ 0.001). The same statistical analysis could not performed for the resectable cases due to the low number of cases in each in sub-group. KRAS mutation validation in primary tissue was only possible in 8 out of 54 (15%) samples due to problems associated with primary tissue availability and a high non-neoplastic cell content of primary tumors.

The overall DNA yield with the QIAamp circulating nucleic acid kit was higher compared to the Maxwell^®^ RSC kit (Appendix A), although there was a higher level of genomic DNA contamination. Thus, this method was used for subsequent sequencing analyses after re-purification of fragments within the expected range of cfDNA (250 bp). Panel sequencing of 15 genes commonly mutated in primary tumors was performed in order to determine the spectrum of somatic mutations (other than *KRAS*) in H/FPC cases. A total of 3 out of 21 (15%) of H/FPC cases were positive for a *KRAS* mutation (p. Gly12Arg) (Figure 3) and BEAMing data were available for 17 of these cases. The *KRAS* status via BEAMing and sequencing was consistent in 16 cases (3 *KRAS* positive and 14 *KRAS* negative) and a *KRAS* mutation was detected by BEAMing and not by sequencing in one case. This is likely due to the lower sensitivity of sequencing for mutation detection compared to BEAMing. The overall frequency of TP53 mutations in H/FPC cases was 12 out of 21 (57%). Furthermore, mutations in *AKT, ERBB2, KIT,* and *PDGFRA* were also detected in *KRAS* negative H/FPC cases (Figure 3 and Table 2). Sequencing and BEAMing data were available for 7 sporadic cases. The presence of a *KRAS* mutation was confirmed by sequencing in 3 of 4 sporadic cases, again this is likely due to the lower sensitivity of sequencing analysis. Of the 3 *KRAS* negative cases determined by BEAMing, one was negative for mutations via sequencing, one harbored mutations in *PDGFRA* and *TP53,* and the other case had mutations in *PIK3CA, KIT, BRAF* and *ERBB2* (Table 2). The KRAS mutation frequency in these 7 sporadic cases was 43%. However, it is important to note that as 3 cases were specifically selected as they were negative for a KRAS mutation via BEAMing and 4 *KRAS* positive cases were included for comparison. Thus, there is a selection bias that is reflected in the KRAS mutation frequency.

Overall, 20 H/FPC and 7 sporadic cases were analyzed by sequencing analysis. Then, 3 out of 21 (14%) of H/FPC were positive for a KRAS mutation. All KRAS mutations were Gly12Asp. One case only had a KRAS mutation, another KRAS and TP53 (His193Arg), and another KRAS, TP53 (Tyr220Cys), and ERBB2 (Asn850Ser). Of the 3 sporadic cases negative for a KRAS mutation via BEAMing, one had a mutation in TP53 and PDGFR, another in PIK3CA, KIT, BRAF, and ERBB2, and the final case was mutation negative via sequencing.

According to the COSMIC database, *KRAS* and *TP53* are the most commonly mutated genes in PDAC, 64 and 47%, respectively. The *KRAS* mutations identified were known pathogenic mutations that have been previously described in COSMIC. The most frequent *KRAS* mutations found by sequencing were c.35G>A G12D and c.34G>C G12R, which are among the most frequent mutations reported in COSMIC (G12D (47%), G12V (31%), and G12R (13%)). The TP53 variant p.(His214Tyr) was identified in 5 H/FPC cases and had a high pathogenicity score (FATHMM prediction (Pathogenic (score 1.00)) but was not found in COSMIC [23].

Survival analysis was performed with hereditary or familial (H/FPC) cases and sporadic cases, which was corrected for sex, age, and stage at diagnosis and 1st line therapy. Hereditary or familial cases had a significantly longer median overall survival compared to sporadic cases; 10.2 vs. 21.7 months, respectively, (*p* ≤ 0.001) (Figure 4). The Hazard Ratio for sporadic cases was 2.4 (*p* ≤ 0.001) (Appendix A), indicating that sporadic cases have a poorer overall survival, independently of stage at diagnosis and 1st line treatment.

Follow-up was censored at 5 years, adjusted for sex, age and disease stage at diagnosis, and first line treatment. The median OS for H/FPC cases was 21.47 months (0.37–60) and 10.4 months (0.57–60) for sporadic cases, which was statistically significant. The HR for sporadic cases was 2.4. Sporadic PDAC: *N* = 102 and H/FPC: *N* = 35.

## 4. Discussion

Circulating-free DNA concentration in plasma or serum has been shown to be a surrogate marker of tumor burden, survival, and recurrence in various tumor types, such as lymphoma [24], lung [25,26], prostate [27] melanoma [28], and colon [29,30]. In this study, the levels of cfDNA correlated with disease status, with higher levels found in PDAC cases vs. healthy individuals or patients with pancreatic cysts. There was a reduction in the total cfDNA levels after resection of the primary tumor, which also supports the notion that cfDNA levels may be indicative of tumor burden. Furthermore, PDAC cases with a cfDNA high concentration at diagnosis had a significantly poorer OS compared to cases with low and medium levels. The lack of primary PDAC tissue of sufficient quality and quantity is a significant problem due to the low resection rate and contamination with the stroma cells, among other reasons. Thus, the liquid biopsy is the most feasible approach for molecular studies in PDAC and we show here that molecular analysis in cfDNA is a good substitute for primary tissue. We attempted to validate *KRAS* mutations in primary tissue but were successful in only 15% of cases, which is a recurring problem in many PDAC somatic profiling studies [31,32].

Mutant *KRAS* was more frequently detected in cfDNA from sporadic cases compared to hereditary or familial cases (70 vs. 16%, respectively). This is the first time that a difference in *KRAS* mutation status frequency has been shown between sporadic and hereditary or familial cases. Previous studies have reported that the frequency of KRAS mutations is similar between familial and sporadic cases [33,34]. However, as they reply on primary tumor samples, they have some limitations, such as sample size and case selection bias. One study reported a similar frequency of *KRAS,* TP53, *CDKN2A*, and *SMAD4* mutations in sporadic and familial cases [33]. However, this study only included primary tumor-derived cell lines from 16 patients, thus there is an important sample size limitation. A second study included 39 cases and also showed a similar somatic profile between somatic and familial cases [34]. However, this study excluded cases with a known pathogenic germline mutation that predisposes them to develop PDAC. Thus, there is an important selection bias, as the cases that we hypothesize will lack KRAS somatic mutations in the primary tumor were excluded. Thus, these data cannot be directly compared with the data presented here.

Although it could be expected that the majority of sporadic cases would be positive for *KRAS* mutation, we did not detect a *KRAS* mutation in all cases. This may be due to the low cfDNA yield which appears to be dependent on disease stage. Furthermore, some patients may not actively shed cfDNA into the blood stream and, therefore, it is almost impossible to detect tumor cfDNA in these individuals. However, there was no statistically significant difference in the median cfDNA levels between sporadic and hereditary or familial cases that may account for this difference in *KRAS* detection frequency. Thus, suggesting that the somatic *KRAS* mutation frequency differs between sporadic and hereditary for familial cases. Furthermore, the *KRAS* positivity rate in the sporadic cases is consistent with previous studies based on cfDNA that report values of 70 and 62.8%. Importantly, we show for the first time that hereditary or familial cases have a longer OS survival compared to sporadic cases, even though they are treated the same regimens in the clinic. Furthermore, hereditary or familial cases were diagnosed at a significantly younger age compared with sporadic cases, 70 vs. 65.5 years, respectively. However, since the survival analysis was corrected for age and disease stage at diagnosis and 1st line adjuvant treatment, the difference in survival may be due to a distinct molecular somatic profile of H/FPC cases compared to sporadic cases.

The Maxwell system was used for BEAMing analysis as the magnetic beads-based approach has been shown to be superior to silica membrane-based methods such as the QIAamp kit [35]. In fact, we showed that Maxwell system favored the extraction of small sized fragments which correspond to cfDNA, with less genomic DNA contamination. Thus, we believe that this method is more appropriate for PCR based methods, such as BEAMing to avoid the amplification of high molecular weight genomic DNA from normal cells. The QIAamp Kit produces the highest yield of total cfDNA compared to other commercially available kits and was therefore used for sequencing analysis [36], which is at least 10 fold less sensitive than BEAMing. The probability of finding a low number of cfDNA molecules follows the Poisson distribution and therefore this probability increases with increased sample volume. A minimum of 3.6 ng of cfDNA is needed to obtain a sensitivity of <0.1% and 36 ng are needed to obtain a sensitivity of <0.01% [35]. Therefore, the sensitivity to detect somatic mutations by sequencing analysis is much lower compared to BEAMing.

There are some important limitations with regard to the use of cfDNA as a prognostic marker that should be taken into account. Some reports have shown that cfDNA isolated from serum has a higher integrity than in plasma, although the *KRAS* allelic frequencies were much lower in serum compared to plasma, with a similar sensitivity and specificity [36]. The critical factors that influence the quality and yield of cfDNA are the time from sample extraction to processing and sample collection tube type and a second centrifugation is required to remove lysed white blood cells [36]. Plasma contains many PCR inhibitors such as heparin, hemoglobin, hormones, immunoglobulin G, and lactoferrin [37], which may be overcome for PCR applications by diluting the sample. Although, sequencing technologies are particularly sensitive to inhibitors and, thus, high quality and quantity samples are needed. Clonal hematopoiesis of indeterminate potential (CHIP) can interfere with cfDNA testing results and the parallel analysis of a paired whole blood control to exclude CHIP variants and avoid misdiagnosis has been recommended [38].

In Europe, PDAC patients are all similarly treated in the clinic. Even though it is clear that there are several PDAC sub-types at the genetic and histological level that differ in terms of prognosis and response to therapy [39]. The backbone of chemotherapy in PDAC are cytotoxic agents that target rapidly proliferating cancer cells. These include the FOLFIRINOX scheme (folinic acid, fluorouracil, irinotecan, and oxaliplatin), gemcitabine combined with nab-paclitaxel, capecitabine and gemcitabine or 5FU monotherapy according to disease stage and Eastern Cooperative Oncology Group (ECOG) status [40]. However, PDAC cases with germline and somatic mutations in DNA repair genes (*BRCA1, BRCA2, PALB2, ATM, BAP1, BARD1, BLM, BRIP1, CHEK2, FAM175A, FANCA, FANCC, NBN, RAD50, RAD51, RAD51C*), are more sensitive to platinum agents in first line and PARP inhibitors (e.g., olaparib and rucaparib) as a maintenance treatment [41,42]. Clinical guidelines recommend PDAC gene profiling and The National Comprehensive Cancer Network (NCCN) guidelines suggest clinical trial participation as first line and second line treatment options [40]. As treatment modifies the genomic composition of PDAC, cfDNA is a useful tool to identify resistance mutations, and potential new treatment targets [43].

Low prevalence focal amplifications in druggable oncogenes have been identified in PDAC such as *ERBB2, MET, FGFR1, CDK6, PIK3R3,* and *PIK3CA*, which may also be exploited in the clinic to provide alternatives to cytotoxic therapies [44]. We hypothesized that hereditary or familial PDAC cases have a distant somatic mutation spectrum due to the presence of germline pathogenic mutations in DNA repair genes. We found somatic mutations in the genes *KIT, AKT, PDGFRA, MET, PIK3CA* and *BRAF* in *KRAS* negative cases, which were mainly found in hereditary and familial cases. Thus, this subgroup of patients could be candidates for treatment with small molecule tyrosine kinase inhibitors (TKI) that inhibit *KIT* and *PDGFR* (e.g., axitinib and imatinib), new generation isoform-specific PI3K inhibitors that reduce toxicity (e.g., alpelisib), small molecule inhibitors and monoclonal antibodies against the receptor and ligands of *MET, BRAF,* and *EGFR*, among others. However, the efficacy of these treatment strategies must be confirmed in pre-clinical or clinical studies in this sub-group of patients.

We show here for the first time that hereditary or familial cases have a lower *KRAS* mutation frequency compared to sporadic cases. Although the data presented here are preliminary and should be validated in a larger cohort of patients, the observation of KRAS negativity in hereditary and familial cases could have an important impact in the clinic for this patient subgroup. In addition, *KRAS* negative cases harbored somatic mutations in potentially druggable genes that could potentially be exploited in the clinic. Furthermore, they have a longer overall survival, which does not appear to be related to stage at diagnosis or 1st line treatment strategy. The main limitation of this study is the sample size. Hereditary or familial PDAC is not a common entity and thus this factor limited the number of cases that were included. Moreover, only 54 samples were included in the BEAMing analysis and 27 in the sequencing analysis due to economic constraints and 8 cases in primary tissue validation due to sample availability. However, there are some positive aspects of this study that should be highlighted. Firstly, as liquid biopsy samples were used, were able to include localized and advanced PDAC cases in the study, which provides a more representative patient cohort. Secondly, we show that the liquid biopsy is a valid alternative in many PDAC cases to primary tissue samples, due to the advent of new technologies with a high sensitivity and specificity for somatic mutation detection. Finally, this study provides preliminary data to suggest that Hereditary or Familial PDAC have a distinct somatic mutation profile compared to sporadic cases and this should be taken into account in the clinic when defining a treatment strategy.

## 5. Conclusions

cfDNA is a valuable source of genomic information in PDAC cases where primary tissue samples are scarce and is also useful to track the genomic changes induced by treatment and tumor dynamics, for the design of a more personalized treatment. The level of cfDNA in plasma appears to be a prognostic indicator, independently of the detection of tumor specific mutations and appears to be a valid substitute for primary tumor tissue for molecular studies in PDAC. Hereditary or familial and tend to be *KRAS* negative and harbor somatic mutations in TP53 in combination with potentially druggable mutations in genes such as *KIT, AKT, BRAF, PIK3CA,* and *PDGFR*. However, these preliminary findings must be validated in a larger cohort. Hereditary or familial PDAC cases have a greater overall survival rate, even though they are treated with the same regimens in the clinic as sporadic cases.

## Figures and Tables

**Figure 1 cancers-13-01612-f001:**
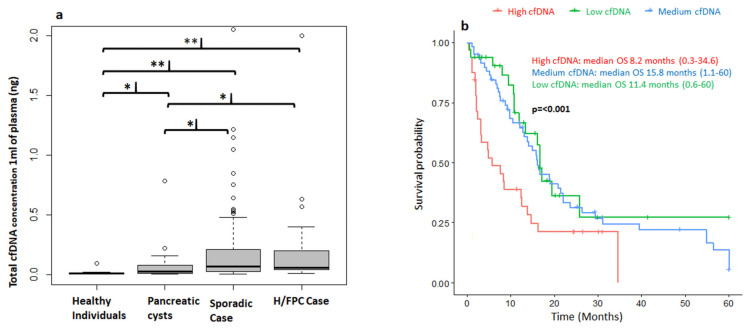
Correlation of circulating free DNA (cfDNA) levels with clinical characteristics. (**a**) The concentration of circulating free DNA (cfDNA) in plasma differentiates between cancer and non-cancer cases. There were significantly higher levels of total cfDNA in hereditary or familial and sporadic pancreatic ductal adenocarcinoma (PDAC) cases compared to healthy controls and patients with pancreatic cysts. ** *p* ≤ 0.01; * *p* ≤ 0.05. (**b**) High levels of cfDNA at diagnosis correlate with a poorer overall survival (OS). Patients were classified into 3 groups: high (>0.2037 ng), medium (≥0.035 ng and ≤0.2037), and low (<0.035 ng) cfDNA levels. Follow-up was censored at 5 years and adjusted for sex, age, and disease stage at diagnosis, sporadic or hereditary and familial, case and first line treatment. Low: *N* = 33; Medium *N* = 67 and High: *N* = 31.

**Figure 2 cancers-13-01612-f002:**
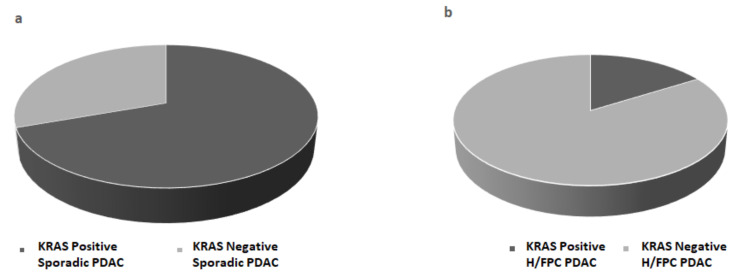
*KRAS* mutation status was determined in plasma from (**a**) sporadic PDAC cases (**b**) hereditary or familial PDAC (H/FPC) cases via BEAMing and mutant *KRAS* was more frequently detected H/FPC cases compared to sporadic cases. BEAMing was performed using cfDNA isolated from 1 mL of plasma from 54 PDAC cases (31 familial cases and 23 sporadic cases). The frequency of mutant KRAS 70% in sporadic cases and 16% in familial cases, which was statistically significant (*p* ≤ 0.001).

**Figure 3 cancers-13-01612-f003:**
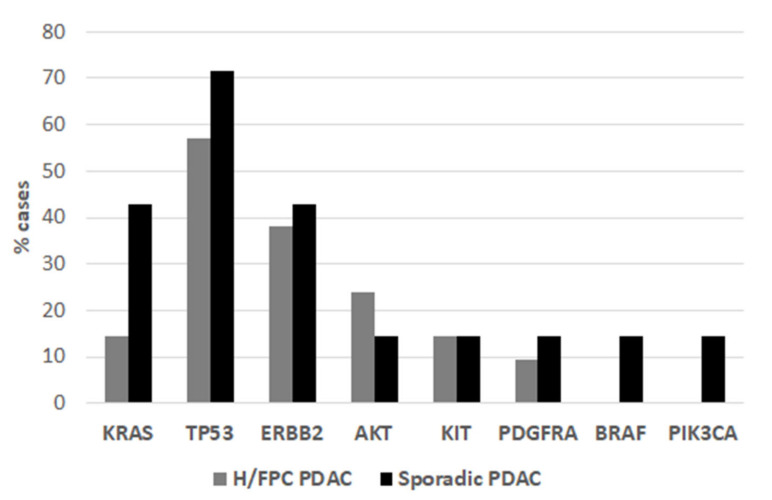
The frequency of somatic mutations in hereditary or familial cases and sporadic was determined by sequencing analysis.

**Figure 4 cancers-13-01612-f004:**
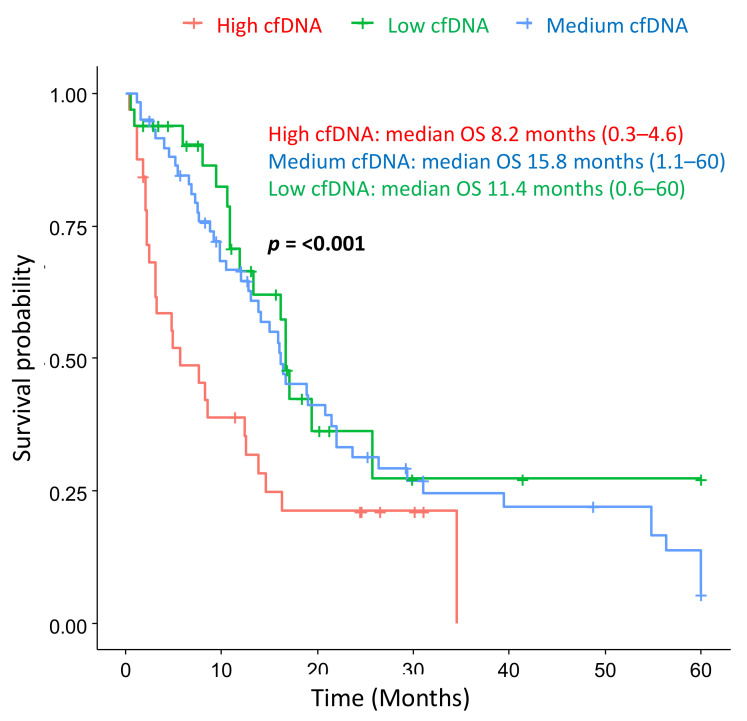
Hereditary or familial PDAC cases have a longer overall survival (OS) compared to sporadic cases.

**Table 1 cancers-13-01612-t001:** Age and sex distribution of the individuals included in the study, according to the different groups analyzed pancreatic ductal adenocarcinoma (PDAC) cases (familial or hereditary (H/FPC) and Sporadic PDAC), pancreatic cysts and healthy controls).

Variable	All PDAC Cases	Hereditary or Familial PDAC	Sporadic	Pancreatic Cysts	Healthy Controls
Male	71	17	54	10	23
Female	74	26	58	19	17
Ratio M:F	0.96	0.65	0.89	0.53	1.35
Median age (range)	69 (29–90)	65.5 (29–84)	70 (31–90)	53 (37–81)	39 (18–63)

**Table 2 cancers-13-01612-t002:** Summary of the somatic mutations detected in H/FPC and sporadic cases by sequencing analysis.

Case	Somatic Mutations Detected by Sequencing	KRAS Status by BEAMing
H/FPC	*KIT*: p.(Glu640Lys) and *TP53*: p.(His214Tyr) and *ERBB2*: p.(Asn850Ser)	md
	*KRAS*: p.(Gly12Arg) and *TP53*: p.(His193Arg)	md
	*KRAS*: p.(Gly12Asp)	md
	*KRAS*:p.(Gly12Asp) and *TP53*:p.(Tyr220Cys) and *ERBB2*:p.(Asn850Ser)	md
	*AKT*: p.(Lys39Asn) and *TP53*: p.(Pro250His) and *ERBB2*: p.(Arg849Gln)	nmd
	*ERBB2*: p.(Ala710Val)	nmd
	*ERBB2*: p.(Cys560Phe) and *AKT*: p.(Thr34Asn) and *TP53*: p.(Pro250His)	nmd
	*PDGFR*A: p.(Ala840Thr) and *AKT*: p.(Thr34Asn) and *TP53*:p.(Pro250His)	nmd
	*TP53* p.(Ser215Ile) and *PDGFR*A p.(Ala840Val) and *KIT* p.(Tyr545Phe)	nmd
	*TP53*: p.(Glu171Gly) and *AKT*: p.(Thr34Asn)	nmd
	*TP53*: p.(His214Tyr)	nmd
	*TP53*: p.(His214Tyr)	nmd
	*TP53*:p.(His214Tyr) and *ERBB2*:p.(Tyr735Cys)	nmd
	*TP53*:p.(His214Tyr) and *KIT*:p.(Glu640Lys) and *ERBB2*:p.(Tyr735Cys) and *ERBB2*:p.(Asn850Ser)	nmd
	NEG	nmd
	NEG	nmd
	NEG	nmd
	NEG	nmd
	*ERBB2*:p.(Leu715Arg) and *AKT*:p.(Pro24Ser)	not tested
	NEG	not tested
	NEG	not tested
Sporadic	*KRAS*: p.(Gly12Asp) and *TP53*: p.(Gly266Val)	md
	*KRAS* p.(Gly12Asp) and *TP53* p.(Gly266Val)	md
	*KRAS*: p.(Gly12Asp) and *TP53*: p.(Gly244Ser) and *AKT*: p.(Arg23Trp) and *ERBB2*: p.(Gly732Asp)	md
	*TP53*: p.(Pro177Arg) and *ERBB2*: p.(Pro1130His)	md
	*PDGFR*A p.(Ala820Val) and *TP53* p.(Pro223Ser)	nmd
	*PIK3CA*: p.(Ala995Asp) and *KIT*: p.(Pro832Ser) and *BRAF*:p.(His585Tyr) and *ERBB2*: p.(Pro761Thr)	nmd
	NEG	nmd

NEG: negative for somatic mutations by sequencing analysis. md: KRAS mutation detected by BEAMing. nmd: no KRAS mutation detected by BEAMing.

## Data Availability

The data presented in this study are available on request from the corresponding author. The data are not yet publicly available.

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
