# Peer review of "Somatic Mutation Profiling in the Liquid Biopsy and Clinical Analysis of Hereditary and Familial Pancreatic Cancer Cases Reveals KRAS Negativity and a Longer Overall Survival"

_cancers, 2021, doi:10.3390/cancers13071612_

Round 1

Reviewer 1 Report

The Authors submitted a very well-revised version of the manuscript and promptly responded to the reviewer's comments. In fact, the authors increased the sound of the manuscript. The manuscript can be accepted for publication.

Reviewer 2 Report

The authors have answered all my consideration. I suggest to collect more cases for expand your database in the future.

This manuscript is a resubmission of an earlier submission. The following is a list of the peer review reports and author responses from that submission.

Round 1

Reviewer 1 Report

The manuscript by Earl et al describes mutational analysis in liquid biopsies from hereditary and familial pancreatic cancer patients. Levels of circulating free DNA are also correlated with clinical/pathological characteristics. A comparison is made with the mutational profile of sporadic pancreatic cancer.

The findings in this manuscript are important since we do not yet have solid knowledge on the mutational profile of hereditary and familial pancreatic cancer

There are a few remarks:

Major remarks:

- I would like to see the major conclusion on lower prevalence of KRAS mutations in the H/FPC confirmed on the primary resected tumor sample of H/FPC.

- It would be informative to gather results on mutation analysis of the pancreatic cysts.

Minor remarks:

- Are cfDNA levels be related to tumor sizeprior to surgical resection?

- The discussion on personalised medicine based on potentially druggable mutations (see also title) should be put in perspective of the -in my opnion- limited successes of basket trials.

Reviewer 2 Report

This is an interesting result to point our the incidence of KRAS mutation in hereditary or familial cases is significantly less compared to that in sporadic cases via BEAMing method. I have some suggestions to be explored:

Major:

  1. Since only 54 cases for KRAS mutation analysis, do patient charactersitics show any difference between 31 H/FPC cases and 23 sporadic cases? Do any other factors, such as initial stages, pathology characteristics or BRCA 1/2 germline status lead to the unbalanced distribution of KRAS mutations between H/FPC and sporadic populations?
  2. How do you pick up genes for sequencing panel? The panel  includes only 15 genes, which are not at all  involved in pancreatic cancer frequently (except TP53 or MET).
  3. Figure 3 showed the frequency of somatic mutations in H/FPC cases only. In addition to KRAS mutation, it's better to show the incidence of other somtaitc mutations in sporadic cases for comparison.
  4. The discussion part stated that TKIs to inhibit  KIT and PDGFR could be considered in H/FPC PC. What kinds of KIT or PDGFRA alternations are validated in 31 H/FPC PC cases. It's better to show all genetic alternations as the heatmap figure.
  5. What 's authors' explanation for better OS in H/FPC cases that that in sporadic cases?
  6. Minor: gentetic name should be in italics.

Reviewer 3 Report

The authors presented an interesting paper focusing on the analysis of KRas mutations in liquid biopsy in PDAC patients. The content of the manuscript appears to be very important as the authors identified a different Kras mutation index in sporadic versus familial PDACs. The data is very interesting and its use seems to be important.

However, some important points regarding both the methodology, the PDAC patient cohort and the interpretation of the data should be implemented. We invite the Authors to take a look at the list of key comments below:

1) A table summarizing the histopathological data of PDAC patients should be reported. How many stage IV patients were enrolled?

2) Authors should check KRas status in a sub-cohort of PDAC patients (in both groups), looking for genetic evaluation in the primary tumor and match the data with cfDNA analyzes.

3) In the manuscript the role and usefulness of patients with pancreatic cysts and IPMN is not very well explained. Furthermore, the authors did not specify: what types of IPMN do they have? What types of pancreatic cysts do they have? How did the authors assess the extent of these injuries? Had the authors subjected PDAC patients to surgical resection for benign lesions?

4) The Graph of figure 2 reported the all types sporadic PDAC patients, including: Resectable, Locally Advanced and Metastatic PDAC patients. In our personal opinion these three groups should analyzed separately.

5) Kras analyses. We would like to point out the major issues concerning the methodologies used for KRas mutation analyses.

-The total frequency of KRas mutation seems to be very Low with respect to the ratio reported in others papers.

-The authors identified only one type of KRas mutation (G12D and G12R, rarerly). Why the authors didn't find the H61Q mutations?

-Did the authors used some positive controls (i.e. PDAC cell lines) in order to set their methodologies?

6) Finally, the authors conclude that different KRas mutational profile between Sporadic and Hereditary PDAC are present in this study. Did the authors tought that KRas is not the driving oncogene in H-PDAC progression? If yes, Why the authors didn't stress this concept? Are there other genes associate with neoplastic progression in H-PDAC patients?

Round 2

Reviewer 1 Report

The authors have addressed my comments and the manuscript has significantly improved. I believe it can be accepted in its current form.